# Comparison between the persistence of post COVID-19 symptoms on critical patients requiring invasive mechanical ventilation and non-critical patients

Irene Irisson-Mora[1☯¤]*, Angélica M. Salgado-Cordero[2☯¤], Estefanía Reyes-Varón[2☯¤], Daniela J. Cataneo-Piña[3☯¤], Mónica Fernández-Sánchez[4¤‡], Ivette Buendía-Roldán[5¤‡], Miguel A. Salazar-Lezama[2¤‡]*, on behalf of the Occupational Health and Preventive Medicine Consortium[¶]

1 Department of Medicine, Division of Endocrinology, Instituto Nacional de Enfermedades Respiratorias, Ismael Cosío Villegas, Mexico City, Mexico, 2 Department of Occupational Health and Preventive Medicine, Instituto Nacional de Enfermedades Respiratorias, Ismael Cosío Villegas, Mexico City, Mexico, 3 Department of Medicine, Division of Geriatrics, Instituto Nacional de Enfermedades Respiratorias, Ismael Cosío Villegas, Mexico City, Mexico, 4 Department of Infectious Diseases Research Center (CIENI), Division of Dermatology, Instituto Nacional de Enfermedades Respiratorias, Ismael Cosío Villegas, Mexico City, Mexico, 5 Department of Interstitial Lung Diseases, Division of Pulmonary Medicine, Instituto Nacional de Enfermedades Respiratorias, Ismael Cosío Villegas, Mexico City, Mexico

☯ These authors contributed equally to this work.
¤ Current address: Instituto Nacional de Enfermedades Respiratorias (INER), Ismael Cosío Villegas, Tlalpan, Mexico City, Mexico
‡ MFS, IBR, and MASL also contributed equally to this work.
¶ Occupational Health and Preventive Medicine Consortium is provided in the Acknowledgments.
* miguelsalazar02@gmail.com (MSL); irene.irisson@iner.gob.mx (IIM)

## Abstract

### Background

During follow-up, patients severely affected by coronavirus disease 2019 (COVID-19) requiring invasive mechanical ventilation (IMV), show symptoms of Post-Intensive Care Syndrome (PICS) such as cognitive impairment, psychological disability, and neuromuscular deconditioning. In COVID-19 pandemic, it is a priority to develop multidisciplinary post-acute care services to address the long-term multisystemic impact of COVID-19.

### Research question

Which are the most relevant multisystemic sequelae in severe post-COVID-19 patients?

### Study design and methods

Observational chart review study that included adult patients discharged from a referral hospital for respiratory diseases in Mexico after recovering from severe COVID-19 disease from December 23, 2020, to April 24, 2021. Data were collected from 280 of 612 potentially eligible patients to evaluate persistent symptoms and compare sequelae in patients who required intubation, using a standardized questionnaire of symptoms, in addition to findings

**Data Availability Statement:** All relevant data are within the manuscript and its Supporting Information files.

**Funding:** The authors received no specific funding for this work.

**Competing interests:** The authors have declared that no competing interests exist.

reported during the face-to-face health assessment. Univariable and multivariate analyses were performed for the association among the requirement of IMV and the long-term persistence of symptoms.

## Results

280 patients were included. The median age was 55 (range, 19 to 86) years, and 152 (54.3%) were men. The mean length of hospital stay was 19 (SD, 14.1) days. During hospitalization 168 (60%) participants received IMV.

A large proportion of these patients reported fatigue (38.7%), paresthesia (35.1%), dyspnea (32.7%) and headache (28%); meanwhile only 3 (1.8%) of them were asymptomatic. Patients who required intubation were more likely to have neuropsychiatric (67.3% vs 55.4%; OR, 1.79 [95% CI, 1.08 to 2.97]) and musculoskeletal involvement (38.7% vs. 25.9%; OR, 1.92 [95% CI, 1.12 to 3.27]), adjusted for age,sex and hospitalization time.

## Interpretation

The proportion of patients requiring intubation was 60%, reporting persistent symptoms in 98% of them. Neuropsychiatric and musculoskeletal symptoms were the most predominant symptoms in these patients, with a significant difference. Post-COVID-19 syndrome is a frequent problem in patients who required IVM. Physicians in ICU and in care of COVID-19 patients should be aware of this syndrome in order to avoid more complications.

## Introduction

By the end of the first half of 2021, coronavirus disease 2019 (COVID-19) has affected more than 178 million people, resulting in almost 4 million deaths world wide [1]. The risk of severe disease [2] and the rate of invasive mechanical ventilation (IMV) reported in countries such as China and the United States goes from 29.1 [3] to 89.9% [4]. Despite high mortality in patients with severe disease, there is a large number of surviving patients who will have to cope with multisystemic sequelae [5, 6]. The lung damage, severe hypoxia, coagulation and inflammatory system abnormalities [7] can lead to cardiac, [8] neurological, [9] kidney and liver injuries [10]. Together, these multisystemic damage increases the odds of having multiple sequelae with variable duration.

As cases of severe COVID-19 survivors increase, also does the number of patients with persistent symptoms, although they have a negative polymerase chain reaction (PCR) test for SARS CoV-2; this syndrome has been called post-COVID-19 syndrome [11]. Some studies have explored these persistent symptoms; in a French longitudinal study, 68% of patients had persistent symptoms at six months of follow up, being fatigue and dyspnea the most frequently reported, in about one third of patients [12–14], followed by pain and psychological distress [15, 16].

These studies clearly indicate that there is a broad extension of post-COVID-19 symptoms, which duration is variable. Most research has included ambulatory patients, with a small proportion of critically ill and mechanical ventilated patients. Our hospital, is a third level respiratory referral center, consequently, receiving the most complicated patients in the country; a multidisciplinary team evaluates all the patients that are discharged from hospitalization. In this single-center study, we assess the persistent symptoms in those patients with severe COVID-19, more specifically on those who needed IMV, so far lacking in the literature.

## Study design and methods

This is an observational, transversal and descriptive study characterized by physical and electronic chart review of the post-COVID-19 cohort patients seen at the National Respiratory Diseases Institute in Mexico City. We included all patients discharged from the hospital with diagnosis of COVID-19 disease from December 23, 2020, to April 24, 2021, where most of the patients were admitted under criteria of severe infection with SARS CoV-2 as defined by the National Institutes of Health (SpO2 > 94%, ratio between arterial partial pressure of oxygen and fraction of inspired oxygen (PaO 2/FiO2) <300 mm Hg, respiratory rate >30 breaths/min [17].

Our Institute is a referral hospital and since patients come from all around the country and due mostly to economic issues, it is impossible for the majority of them to return to the hospital for a follow up check-up (post-COVID-19 follow up). Due to this limitation, all COVID-19 discharged patients undergo a telephone assessment up to 6 months after discharge where a systematic questionnaire of potentially persistent symptoms related to COVID-19 is carried out (providing informed verbal consent during this call). The questionnaire applied is a non-validated test, developed by attending physicians in charge of the post-COVID-19 clinic in our Institution. Patients with persistent symptoms or any complain are asked to come back to the hospital, where detailed physical exam and if necessarily other complementary examinations and tests are performed. All these data are collected in an electronic data system. The Institutional Ethics Committee approved this study (COD-C22-21).

All patients included in this study had been hospitalized with diagnosis of COVID-19 infection by positive polymerase chain reaction (PCR) test, from december 23, 2020, to april 24, 2021; time from discharged ranged between 2 weeks and 6 months. The cut off value for prolonged hospitalization was determined as hospital stay longer than 14 days. No COVID-19 virus variant was routinely performed during hospitalization and was not consider in this study.

Thus, 612 patients were potentially eligible, of which 16 died prior to possible follow-up, 32 did not answer the phone call, 22 were hospitalized due to another diagnosis, three required hospital re-admission, two were transferred to another unit where they continued their follow-up, 58 requested telemedicine follow-up due to the inability to attend the assessment in person, 4 voluntary discharges, 10 refused a follow-up visit, 14 did not attend the day of their scheduled follow-up appointment, 167 already had post-COVID-19 follow-up care at another clinic or institution, and finally four patients were excluded, since they had more than 24 weeks from their hospital discharge; so, 280 patients were included in this study as mentioned in Fig 1.

As mentioned previously, all discharged patients underwent a telephone assessment up to 6 months after discharge where a systematic questionnaire of potentially persistent symptoms related to COVID-19 was carried out, and if applicable they were asked to attend a medical evaluation at the Institute where a detailed physical examination and complementary examinations were performed (including lab tests and CT Torax scan if necessarily).

Chart review was performed, and we classified the persistent post-COVID-19 signs and symptoms in 8 different domains accordingly to the organ or system affected.: [1] Respiratory symptoms, including dyspnea, chest pain, cough, odynophagia, rhinorrhea, wheezing, abnormalities of the vocal cords (hoarseness, cord paralysis) and tracheal stenosis [2]. Cardiovascular symptoms such as clubbing, rhythm disorders (palpitations, tachycardia, and bradycardia), heart murmur, pericarditis, elevated blood pressure, lower limb edema and deep vein thrombosis [3]. General symptoms, like fever, dizziness, fatigue, asthenia, adynamia, sweating, itching, cold sensitivity, and tremors [4]. Musculoskeletal symptoms as well as lack of sensation,

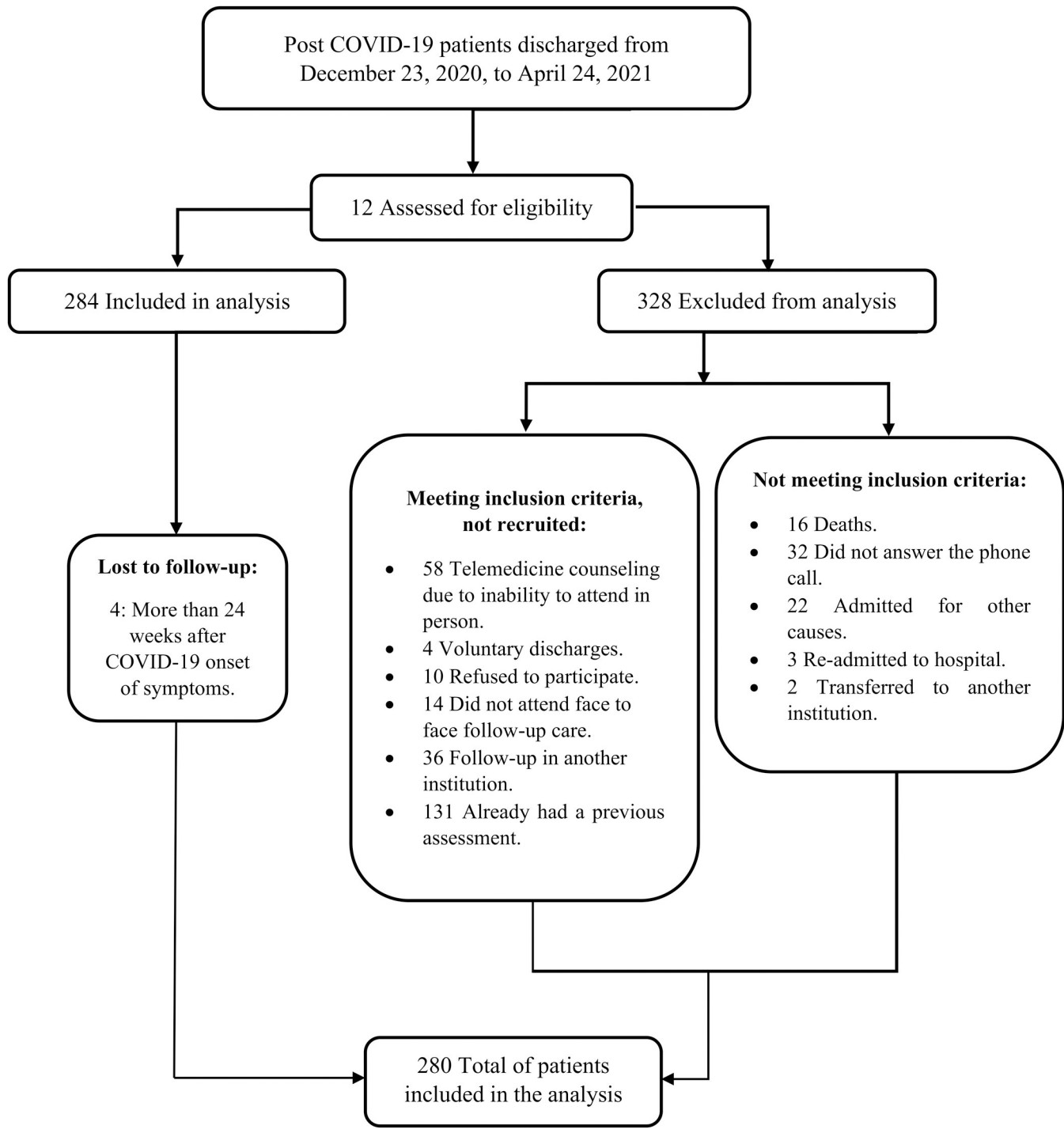

**Fig 1. Flow diagram of the study design.** In this flow diagram, the quantity of patients who were assessed for eligibility and the exclusion criteria are exposed.

mobility limitations, arthralgias, tendon injuries, myalgias, paresthesia, neuralgia, muscular weakness, low back pain, and contractures [5], Dermatological symptoms, such as hair loss, skin rashes, and pressure ulcers [6]. Neurological and Neuropsychiatric symptoms, such as headache, sleep-wake disorders, cognitive impairments such as lack of concentration and

memory loss, as well as anxiety and/or depression disorders [7], Gastrointestinal symptoms, including nausea, diarrhea, constipation, abdominal pain, and lack of appetite [8]. Audiologic symptoms, including tinnitus, earache, hearing loss, and vertigo. The impact of each domain on the daily function of the participants has been reported as present or absent.

All data from the physical and electronic charts were collected in an Excel page, which we imported for analysis into a statistical program IBM SPSS (Statistical Package for the Social Sciences) version 25.

Data were presented as measures of central tendency and dispersion. Qualitative variables were expressed as frequencies and percentages, while continuous variables are reported as means ± standard deviations (SD) or as medians with interquartile ranges (IQR), depending on their distribution.

An association analysis was performed to compare both groups (according to IMV requirements) using the Chi-squared test (X2), or Fisher's exact test and Student's T test for univariable analyses. Afterwards, multivariate logistic regression models were performed to yield adjusted odds ratio (AORs) with 95% confidence intervals (CI), to evaluate the association between variables that in the univariable analysis showed a statistically significant difference, considering significant P values under 0.05.

## Results

From 280 patients included in the study, the median age was 55 (range, 19 to 86) years, of which 180 (64.3%) were under 60 years of age; 152 (54.3%) were males and 128 (45.7%) females. The average of hospital stay length and time since the onset of symptoms through follow-up period were 19 (± 14.1) days and 10 (± 3.73) weeks, respectively.

Upon admission, mean oxygen saturation (measured by pulse oximetry) was 73.45% ± 13.3% (range, 20 to 97%)

During hospitalization, two-third (168 of 280, 60%) required IMV for a mean time of 14.93 ± 8.74 days and 112 (40%) non required IMV; 31(11.1%) required tracheostomy, and 11 (3.9%) required gastrostomy. The most common comorbidities were obesity and overweigh (81.1%), hypertension (41.4%) and diabetes mellitus (29.3%); while only 35 (12.5%) were previously healthy. Demographic and IMV-related characteristics are shown in Table 1.

At the time of evaluation, up to six months after discharged, only 3 of the 168 intubated patients (1.8%) were asymptomatic and 114 (67.9%) reported more than 3 persistent symptoms. Fatigue (41.4%) was the most common symptom, followed by headache (27.5%), paresthesias and neuralgia (26.1%); while, in the multisystemic evaluation, respiratory sequelae (65%, 182 of 280), followed by neurological and neuropsychiatric sequelae (62.5%, 175 of 280) were the most frequent symptoms (Table 2); describing dyspnea in 35.4% (99 of 280), chest pain in 21.1% (59 of 280), cough in 17.5% (49 of 280), sleep-wake disorders in 12.1% (34 of 280), as well as anxiety and depression disorder in 10.4% (29 of 280) and abnormalities of vocal cords in 9.6% (27 of 280) of the patients.

This was followed by general symptoms (54.3%, 152 of 280) and musculoskeletal sequelae (33.6%, 94 of 280); as prior mentioned fatigue, paresthesia/neuralgia in 26.1% (73 of 280), myalgia in 21.1% (59 of 280), joint pain in 11.8% (33 of 280), mobility limitations 8.6% (24 of 280), tremor in 7.9% (22 of 280), adynamia 7.1% (20 of 280), asthenia 6.4% (18 of 280) and muscular weakness in 5% (14 of 280) (Fig 2).

Some less frequently reported sequelae were cardiovascular (18.6%, 52 of 280), gastrointestinal 13.6% (38 of 280) and dermatological symptoms (16.8%, 47 of 280); such as rhythm disorders (10.7%, 30 of 280), lower limb edema (5.4%, 15 of 280), abdominal pain (2.5%, 7 of 280), constipation (2.1%, 6 of 280), hair loss (10.7%, 30 of 280) and pressure ulcers (5.7%, 16 of 280);

**Table 1. Mechanical ventilation (univariable analyses).** Demographic and clinical features of a group of post COVID-19 patients associated to invasive.

| Variable | TotalN (%)[a] N = 280 | Invasive Mechanical Ventilation N (%) | | P value |
|---|---|---|---|---|
| | | Present N = 168 (60) | Absent N = 112 (40) | |
| >60 years old | 100 (35.7) | 56 (33.3) | 44 (39.3) | 0.309 |
| Male sex | 152 (54.3) | 101 (60.1) | 51 (45.5) | 0.016[d] |
| Comorbidities | | | | |
| Hypertension | 116 (41.4) | 72 (42.9) | 44 (39.3) | 0.552 |
| Diabetes | 82 (29.3) | 49 (29.2) | 33 (29.5) | 0.957 |
| Obesity and/or Overweight | 227 (81.1) | 134 (79.8) | 93(83.0) | 0.493 |
| Dyslipidemia | 17 (6.1) | 10 (6) | 7 (6.3) | 0.919 |
| Hypothyroidism | 16 (5.7) | 5 (3) | 11 (9.8) | **0.016[d]** |
| Heart disease | 7 (2.5) | 5 (3) | 2 (1.8) | 0.532 |
| History of lung disease | 10 (3.6) | 1 (0.6) | 9 (8) | **0.001[b, d]** |
| Previously healthy | 35 (12.5) | 19 (11.3) | 16 (14.3) | 0.461 |
| Risk factors | | | | |
| Smoking | 110 (39.3) | 69 (41.1) | 41 (36.3) | 0.454 |
| Biomass exposure | 46 (16.4) | 23 (13.7) | 23 (20.5) | 0.130 |
| Alcoholism | 23 (8.2) | 17 (10.1) | 6 (5.4) | 0.155 |
| Clinical presentation | | | | |
| ≥3 symptoms | 184 (65.7) | 114 (67.9) | 70 (62.5) | 0.355 |
| 1–2 symptoms | 90 (32.1) | 51 (30.4) | 39 (34.8) | 0.433 |
| Asymptomatic | 6 (2.1) | 3 (1.8) | 3 (2.7) | 0.680 |
| Sequelae | | | | |
| Respiratory symptoms | 182 (65) | 103 (61.3) | 79 (70.5) | 0.103 |
| Cardiovascular symptoms | 52 (18.6) | 33 (19.6) | 19 (17) | 0.572 |
| General symptoms | 152 (54.3) | 90 (53.6) | 62 (55.4) | 0.769 |
| Musculoskeletal symptoms | 94 (33.6) | 65 (38.7) | 29 (25.9) | **0.026 [b, d]** |
| Dermatological symptoms | 47 (16.8) | 34 (20.2) | 13 (11.6) | 0.058 |
| Neurological and neuropsychiatric symptoms | 175 (62.5) | 113 (67.3) | 62 (55.4) | **0.044 [b, d]** |
| Gastrointestinal symptoms | 38 (13.6) | 27 (16.1) | 11 (9.8) | 0.135 |
| Audiologic symptoms | 10 (3.6) | 7 (4.2) | 3 (2.7) | 0.511 |

[a] Data are presented as N (%).

[b] Fisher Exact Test was used to report the p value, otherwise Chi squared test was performed.

[c] COVID-19 = Coronavirus disease 19.

[d] P values in bold have statistical significance.

and finally audiologic symptoms with only 3.6% in 10 of 280 patients, among them earache (2.1%, 6 of 280) and tinnitus (1.8%, 5 of 280).

## Univariable analyses by gender

Men had a higher tendency to become intubated (60.1 vs 45.5%, P = 0.016), had a longer hospitalization time (46.1 vs 32.8%, P = 0.028), more risk factors including smoking (48 vs 28.9%, P = 0.001) and alcoholism (13.2 vs 2.3%, P = 0.001), as well as more comorbidities (51.8 vs 48.2%, P = 0.031). On the other hand, women had a higher biomass exposure (24.2 vs 9.9%, P = 0.002), and more respiratory (71.9 vs 59.2, P = 0.032) and dermatological symptoms (25.8 vs 9.2%, P = <0.001) compared to men.

**Table 2. Univariable analyses by gender, comorbidities and prolonged hospitalization time.**

| Gender | | | | |
|---|---|---|---|---|
| Variable | Total N (%) | Women | Men | P value |
| Comorbidities | 245 (87.5) | 118 (48.2) | 127 (51.8) | **0.031** [d] |
| Risk factors | | | | |
| Smoking | 110 (39.3) | 37 (28.9) | 73 (48.0) | **0.001** [d] |
| Biomass exposure | 46 (16.4) | 31 (24.2) | 15 (9.9) | **0.002** [d] |
| Alcoholism | 23 (8.2) | 3 (2.3) | 20 (13.2) | **0.001** [d] |
| Prolonged hospitalization time | 112 (40.0) | 42 (32.8) | 70 (46.1) | **0.028** [d] |
| Sequelae | | | | |
| Respiratory symptoms | 182 (65.0) | 92 (71.9) | 90 (59.2) | **0.032** [d] |
| Dermatological symptoms | 47 (16.8) | 33 (25.8) | 14 (9.2) | **<0.001** [d] |
| Comorbidities | | | | |
| Variable | Total N (%) | Comorbidities | Previously healthy | P value |
| Sequelae | | | | |
| Respiratory symptoms | 182 (65.0) | 165 (67.3) | 17 (48.6) | **0.037** [d] |
| Musculoskeletal symptoms | 94 (33.6) | 88 (35.9) | 6 (17.1) | **0.034** [d] |
| Prolonged hospital time | | | | |
| Variable | Total N (%) | Present | Absent | P value |
| Risk factors | | | | |
| Smoking | 110 (39.3) | 55 (49.1) | 55 (32.7) | **0.009** [d] |
| Biomass exposure | 46 (16.4) | 12 (10.7) | 34 (20.2) | **0.047** [d] |

[d] P values in bold have statistical significance.

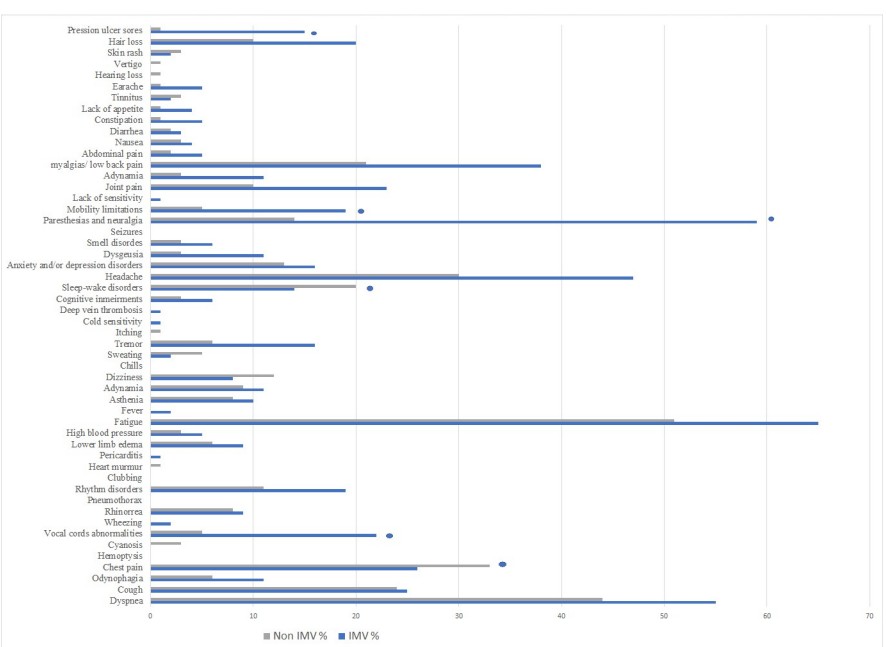

**Fig 2. Post COVID -19 symptoms according to mechanical ventilation requirements.** A comparison of the persistence of symptoms in general, whether a patient required invasive mechanical ventilation or not are shown. In the bar graph patients on invasive mechanical ventilation are represented by the blue column and the non-intubated patients are illustrated by the grey column. * Represents the statistically significant outcomes, according to P values.

### Univariable analyses by comorbidities

Patients with any previously comorbidities reported more respiratory (67.3 vs 48.6%, *P = 0.037*) and musculoskeletal (35.9 vs 17.1%, *P = 0.034*) symptoms compared to previously healthy patients. There were no other significant differences between these two groups.

### Univariable analyses by prolonged hospital time

Patients with longer hospital time had more tendency of smoking (49.1 vs32.7%, *P = 0.009*), and less history of biomass exposure (10.7 vs 20.2%, *P = 0.047*) compared to patients with shorter hospitalization time. There were no other significant differences between these two groups.

### Univariable analyses by IVM

IVM patients had a longer hospitalization time compared with non-intubated patients (59.5 vs 10.7, *P = <0.001*) as well as more musculoskeletal symptoms (38.7 vs 25.9%, *P = 0.026*) and neurological-neuropsychiatric symptoms (67.3 vs 55.4%, *P = 0.044*). On the other hand, non-intubated patients had a higher history of lung disease (8 vs 0.6%, *P = 0.001*) as a well as more hypothyroidism history (9.8 vs 3%, *P = 0.016*) compared to IVM patients. There were no significant differences between intubated and non-intubated patients in terms of other comorbidities, biomass exposure or smoking.

### Multivariable analyses for gender

In a multivariate logistic regression analysis, adjusted for age and prolonged hospitalization time, women were more likely to have dermatological symptoms (25.8% vs 9.2%; OR, 3.464; 95% CI, 1.635–7.336; *P = <0.001*), a higher biomass exposure (24.2% vs 9.9%; OR, 2.401; 95% CI, 1.143–5.045; *P = 0.021*) compared to men. On the other hand, men had higher smoking history (48% vs 28.9%; OR, 0.449; 95% CI, 0.257–0.784; *P = 0.005*), and higher alcoholism history (13.2% vs 2.3%; OR, 0.260; 95% CI, 0.070–0.964; *P = 0.044*) compared to women.

### Univariable analyses by comorbidities

In a multivariate logistic regression analysis, adjusted for age, sex, and prolonged hospitalization time patients with previously comorbidities were more likely to have musculoskeletal symptoms (35.9% vs 17.1%; OR, 0.344; 95% CI, 0.133–0.884; *P = 0.027*), compared to previously healthy patients.

### Multivariable analyses by prolonged hospital time

In a multivariate logistic regression analysis, adjusted for age, sex, and prolonged hospitalization time patients with longer hospital time had more smoking history (49.1% vs 32.7%; OR, 1.867; 95% CI, 1.125–3.099; *P = 0.016*) compared to patients with shorter hospitalization time.

### Multivariable analyses for IVM

In a multivariate logistic regression analysis, adjusted for age, sex, and prolonged hospitalization time, patients in IVM were more likely to have neurological and neuropsychiatric involvement (67.3% vs 55.4%; OR, 1.936; 95% CI, 1.069–3.506; *P = 0.029*), as well as neuromuscular involvement (38.7% vs. 25.9%; OR, 1.946; 95% CI, 1.057–3.591; *P = 0.032*) compared to non-intubated patients (Fig 3 and Table 3).

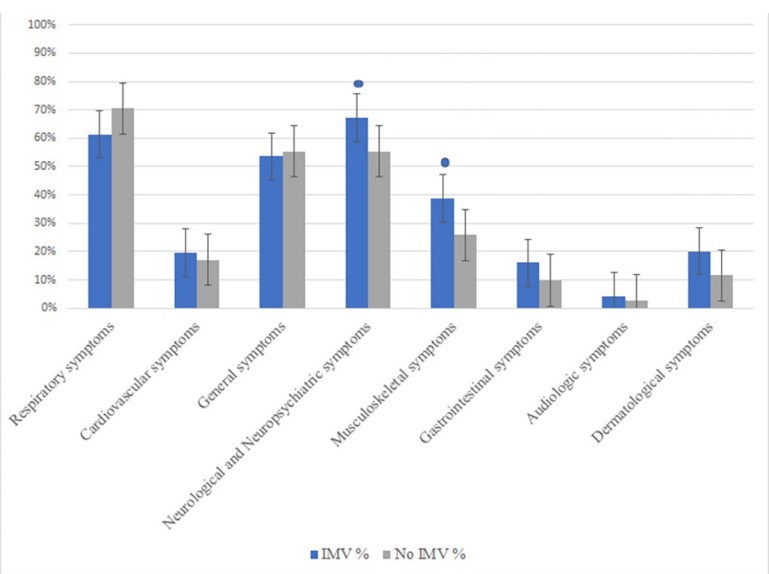

**Fig 3. Post COVID-19 sequelae for IMV and non-IMV patients.** A comparative bar graph of post-covid sequelae, according to the requirements of IMV is represented. In the bar graph patients on invasive mechanical ventilation are represented by the blue column and the non-intubated patients are illustrated by the grey column. * Represents the statistically significant outcomes, according to P values.

## Discussion

Since the beginning of COVID-19, the number of patients with sequelae has increased, making it necessary to direct our attention and efforts to the recovery from these complications. In our study, we sought to identify the most prevalent symptoms of critically ill patients due to

**Table 3. Multivariate logistic regression analysis of gender, comorbidities, prolonged hospitalization time and invasive mechanical ventilation.**

| Variable | Total N (%) | OR | CI | P |
|---|---|---|---|---|
| **Gender** | | | | |
| Risk factors | | | | |
| Smoking | 28.9 vs 48.0 | 0.449 | 0.257–0.784 | **0.005** [d] |
| Biomass exposure | 24.2 vs 9.9 | 2.401 | 1.143–5.045 | **0.021** [d] |
| Alcoholism | 2.3 vs 13.2 | 0.260 | 0.070–0.964 | **0.044** [d] |
| Sequelae | | | | |
| Dermatological symptoms | 25.8 vs 9.2 | 3.464 | 1.635–7.336 | **<0.001**[d] |
| **Comorbidities** | | | | |
| Sequelae | | | | |
| Musculoskeletal symptoms | 35.9 vs 17.1 | 0.344 | 0.133–0.884 | **0.027** [d] |
| **Prolonged hospital time** | | | | |
| Risk factors | | | | |
| Smoking | 49.1 vs 32.7 | 1.867 | 1.125–3.099 | **0.016** [d] |
| **Invasive mechanical ventilation** | | | | |
| Sequelae | | | | |
| Musculoskeletal symptoms | 38.7 vs 25.9 | 1.946 | 1.057–3.591 | **0.032** [d] |
| Neurological and neuropsychiatric symptoms | 67.3 vs 55.4 | 1.936 | 1.069–3.506 | **0.029** [d] |

[d] P values in bold have statistical significance.

COVID-19, 60% of these patients required IMV over the course of the hospitalization, which relates to long-term complications in these specific group of patients.

Our results show that the most prevalent individual symptoms in overall population were fatigue and dyspnea, and grouped by systems, the respiratory system was the most affected, followed by neuropsychiatric symptoms. After analyzing the prevalence of symptoms depending on the mechanical ventilation status, patients who required IMV had more neurological disturbances, specifically neuropathic pain, and more neuropsychiatric symptoms, mainly anxiety, depression and sleeping disorders. A systematic literature review that included 79 studies, which aimed to describe the neurological manifestations of COVID-19, found a high prevalence of symptoms such as olfactory disfunction, ischemic stroke, headache, dizziness encephalitis, neuralgia, and ataxia among COVID-19 patients, but post COVID-19 neurological sequelae have not been previously evaluated [18].

During follow-up, our patients severely affected by COVID-19 who needed IMV showed symptoms of Post Intensive Care Syndrome (PICS) such as cognitive impairment, psychological disability, and neuromuscular deconditioning [19]. COVID-19 critical patients may be predisposed to have a greater prevalence of PICS because of longer periods of mechanical ventilation [20], sedation and the use of steroids and sedative meditation for their treatment [21]. Although PICS is known to be a common syndrome in patients that needed critical care, and there are a significant amount of patients that required these kind of care due to COVID-19, the prevalence of PICS and its definition in patients with COVID-19 is still not yet determined. Such is the case that in 2021 a call has been made to the World Health Organization (WHO) to develop International Classification of Disease Diagnostic Codes for PICS in the "Age of COVID-19" that incentive to care coordination in outpatient settings and to an effective post-acute treatment [22].

The current study found that the patients with pneumopathy had lower risk of IMV requirement than non-invasive ventilation patients, which is consistent with previous publications [23]. Which could be explained by three situations: the lack of recognition, a protective effect by immune response and finally a possible effect of the treatment of these diseases such as use of inhaled corticosteroids alone or in combination with bronchodilators shown on in vitro models [24], that have been able to suppress the replication of coronavirus and the production of cytokines.

The human coronavirus can enter to the central nervous system through the olfactory bulb and the brain receptors for angiotensin-converting enzyme 2 expressed over glial cells and neurons [25]. Findings of previous research over Severe Acute Respiratory Syndrome Coronavirus 1 (SARS-CoV-1) and Middle East Respiratory Syndrome Coronavirus (MERS-COV) show that these respiratory viruses cause severe brainstem damage, contributing to failure of the respiratory centers [26, 27]. Other mechanisms that contribute to the pathophysiology of post-acute COVID-19 include viral direct organ damage and inflammatory damage in response to the acute infection [28]. Although these changes have been observed in different organs and systems, in mechanically ventilated patients, we found a greater amount of neurological, muscular and psychiatric sequelae at long term.

In addition to viral direct damage, SARS-CoV-2 infection triggers an inflammatory storm, with a subsequent break of the blood-brain barrier, which contributes to the neuroinflammatory process [29]. SARS-CoV-2 also produces infectious toxic encephalopathy, which refers to a type of reversible brain dysfunction syndrome caused by systemic toxemia, metabolic disorder and hypoxia [30], leading to mental disorders and delirium [31]. The neuroinflammatory process and persistent hypoxia also damage the hippocampus and cortical areas, causing cognitive function and behavioral alterations [32].

Although the scope of our results is limited to a middle term, the neurological and cognitive impairments may extent to a longer period. This should be assumed according to the recent findings from the UK Biobank study, where COVID-19 patients had a significant loss of grey matter in the left parahypocampal gyrus, the left lateral orbitofrontal cortex and the left insula [33]. In addition to the gray matter decrease, COVID-19 critically ill patients also tend to have more incidence of microbleed at follow up [34], mainly found in the juxtacortical white matter, corpus callosum and internal capsule [35, 36]. The implications of these structural and vascular alterations must be considered at the long term follow up of these patients, therefore cognitive and neurological evaluation should be carried out periodically. Our center provides neurological assessment to identify cognitive impairment, peripheral and central nervous system disorders. According to the identified sequelae, patients are referred to physical and neuropsychological rehabilitation where they receive therapy for up to 6 months until sequelae have improved.

Several reports described a great burden of psychiatric symptoms such as insomnia [37], anxiety, post-traumatic stress symptoms [38] and mood disorders [39] in patients affected by COVID-19 on the acute phase. Chronic psychological distress and post-traumatic stress disorder can develop in patients who survive critical illness and have been reported as a complication of the infection by other coronaviruses such as MERS [40] and SARS [41]. In a systematic review and meta-analysis that included 72 studies that aimed to assess the psychiatric and neuropsychiatric presentation of MERS, SARS and COVID-19, they found that in the post-acute stage, the prevalence of post-traumatic stress disorder was 32.2%, that of depression was 14.9% and that of anxiety disorder was 14.8% [42], data that our study also reports. Beyond the brain damage that could be conferred by viral invasion or indirectly by immune response or medical therapy [43], the psychiatric long-term consequences may arrange from other sources such as social isolation, concerns about other family members infected, the impact of a potentially fatal illness along with physical and cognitive impairments which limit their ability to return to their usual activities [44].

We also found that patients who did not required mechanical ventilation had more sleep-wake disorders compared to intubated patients as mentioned in some publications [45], this has been related to the lack of contact with family and beloved ones in those patients who were aware of the COVID-19 critical situation [46, 47]. These patients underwent to a massive psychological stress, which increased pro-inflammatory markers, in particular protein C and IL-6, contributing to the increase of neuroinflammation [48], and therefore, increasing symptoms of depression associated with sleep disorders, especially insomnia. In our cohort, 10% of patients reported psychological distress, anxiety, depression, or sleep disorders. As part of a multidisciplinary management, our patients receive psychiatric and psychological assessment and treatment, which includes pharmacotherapy and cognitive-behavioral therapy sessions. These interventions provide a significant improvement in these symptoms and are fundamental for patient´s quality of life recovery.

Our study provides a framework to focus the attention on the main affected systems in severely affected COVID-19 patients, mainly on those who received IMV. Some suggested acute-phase interventions for neurological protection include tidal volume minimization in ventilation setting strategies to improve cerebral blood flow and lower cerebral vascular resistance along with less inflammation [49, 50]. The use of some sedative agents such as dexmedetomidine instead of benzodiazepines [51], less use of continuous deep sedation [52, 53], along with and a conservative fluid management [54] have also been proposed as useful interventions to maximize neurological function. Although these can be important neuroprotective strategies, the use of an interdisciplinary team, that endorses early rehabilitation may improve general outcomes in critically ill patients [55].

During follow-up, we also found an association between patients who were assisted with mechanical ventilation and post-intensive care syndrome, mainly with neuromuscular deconditioning, being these sequelae the second most statistically significant, such as paresthesia and neuropathies, and mobility limitations. Concerning to dermatological sequelae, pressure ulcers were significantly present in critical patients, which according to a previous publication from our institute [56], patients who underwent to mechanical ventilation had a considerable association with hospital stay length and obesity as a risk factor, the prevention is the best way to go through by mobilizing patients at least twice per shift and applying lubricants for skin care.

Physical rehabilitation in the intensive care unit (ICU) has been incorporated as a standard care procedure for the prevention of the post intensive care syndrome [57, 58]. Physical rehabilitation improves mobility and muscle strength [59], decreases hospital length of stay [60], reduces hospital readmission [61], and may promote neurogenesis [62], release of neurotrophic factors [63] and increase blood flow [64]. Research on the effectiveness of ICU physical rehabilitation mainly focuses on physical morbidities, and its effect on neurological and psychiatric sequelae on COVID-19 critical patients need to further be studied. Due to the nature of the disease, rehabilitation may not be seen as a priority. The need for personal protective equipment supplies and the risk of contagion may limit the possibility of providing physical therapy among critical patients. Nonetheless, since early and structured rehabilitation improves outcomes for patients requiring prolonged periods of mechanical ventilation [65], the implementation of structured rehabilitation programs which commence on early hospitalization and continue after discharge, is mandatory.

The main limitations of this study include its observational nature where symptoms were self-reported by patients without objective criteria for their determination, which could affect the reliability of symptom prevalence estimates. In addition, being a cross-sectional study, it has the limitation of not being able to discern a specific time line of the symptoms since only one time point was recorded, therefore the interpretation of these findings requires caution. More research is needed to understand the course of the post-COVID-19 syndrome, its underlying mechanisms, and possible rehabilitation programs and treatments that can be implemented.

## Conclusion

In our hospital, where most of the patients were admitted with severe COVID-19 and a large number of them required IMV management, was observed that patients had a greater tendency to present neurological and neuropsychiatric sequelae, in contrast to non-intubated patients. For this reason, we suggest preventive measures such as neuroprotective therapies, early rehabilitation programs, implementation of telemedicine and internet-based mental health interventions, and a system that allows adequate communication between hospitalized patients and their families.

Our study provides a framework to focus the attention on the main affected systems in severely affected COVID-19 patients, mainly on those who receive IMV. We highly recommend the development of a multidisciplinary team specialized in COVID-19 care services in critically ill patients that endorses early rehabilitation programs which commence in hospitalization and continue after discharge, to adequately manage these symptoms and maximize their functional return to their quotidian activities.

## Supporting information

**S1 Appendix. Permissions.** Institutional ethics committee approval dictamen.
(PDF)

**S2 Appendix. Systematized questionnaire of symptoms.**
(PDF)

**S1 File.**
(SAV)

## Acknowledgments

We carried out this project in collaboration with Maribel Mateo Alonso, MD (Head, Outpatient Clinic, Instituto Nacional de Enfermedades Respiratorias, Ismael Cosio Villegas, México City, Mexico) for her invaluable help in coordinating the fieldwork. We are very grateful to Luis E. Morales Bartolo, MD; Nadia Díaz Vázquez, RN; Ana M. Vega-Martínez, RN; Fernando Sosa-Gómez, MD; Karen Jimarez Núñez, Medical practitioner and the rest of the members of Occupational Health and Preventive Medicine Consortium for their support during the study.

### Occupational Health and Preventive Medicine Consortium:

Maribel Mateo-Alonso (Lead author for this group, email: mmateo_75@hotmail.com).
  Luis E. Morales Bartolo.
  Nadia Díaz-Vázquez.
  Ana M. Vega-Martínez.
  Fernando Sosa-Gómez.

## Author Contributions

**Conceptualization:** Irene Irisson-Mora, Angélica M. Salgado-Cordero, Estefanía Reyes-Varón.

**Data curation:** Irene Irisson-Mora, Angélica M. Salgado-Cordero, Estefanía Reyes-Varón.

**Formal analysis:** Irene Irisson-Mora, Angélica M. Salgado-Cordero, Estefanía Reyes-Varón, Daniela J. Cataneo-Piña.

**Investigation:** Irene Irisson-Mora, Angélica M. Salgado-Cordero, Estefanía Reyes-Varón, Daniela J. Cataneo-Piña, Ivette Buendía-Roldán, Miguel A. Salazar-Lezama.

**Methodology:** Irene Irisson-Mora, Angélica M. Salgado-Cordero, Estefanía Reyes-Varón, Daniela J. Cataneo-Piña, Ivette Buendía-Roldán, Miguel A. Salazar-Lezama.

**Project administration:** Mónica Fernández-Sánchez.

**Resources:** Mónica Fernández-Sánchez.

**Supervision:** Mónica Fernández-Sánchez, Ivette Buendía-Roldán, Miguel A. Salazar-Lezama.

**Validation:** Irene Irisson-Mora, Angélica M. Salgado-Cordero, Estefanía Reyes-Varón, Mónica Fernández-Sánchez, Ivette Buendía-Roldán, Miguel A. Salazar-Lezama.

**Visualization:** Irene Irisson-Mora, Angélica M. Salgado-Cordero, Estefanía Reyes-Varón, Daniela J. Cataneo-Piña.

**Writing – original draft:** Irene Irisson-Mora, Angélica M. Salgado-Cordero, Estefanía Reyes-Varón, Daniela J. Cataneo-Piña.

**Writing – review & editing:** Irene Irisson-Mora, Angélica M. Salgado-Cordero, Estefanía Reyes-Varón, Daniela J. Cataneo-Piña.

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
