## [Decision Letter · Decision Letter 0]

17 Mar 2022

PONE-D-21-28914Comparison between the persistence of post COVID-19 symptoms on critical patients requiring invasive mechanical ventilation and non-critical patients.PLOS ONE

Dear Dr. Salazar-Lezama,

Thank you for submitting your manuscript to PLOS ONE. After careful consideration, we feel that it has merit but does not fully meet PLOS ONE’s publication criteria as it currently stands. Therefore, we invite you to submit a revised version of the manuscript that addresses the points raised during the review process.

 Long haul COVID is indeed of rising concern and a significant burden to the healthcare system in future. I commend the authors on researching on this important area of interest. Even thought the study does not add to the existing knowledge it helps to solidify previously proven factors of interest in long COVID. Having said that, there are several concerns that need addressing:

As reviewer 1 suggested, please clarify the documentation of the telephonic consent obtained and if the data was collected before the ethical committee approval. Some of the language need rephrasing to clarify the meaning like the O2 saturation at arrival. Indicate definitions for long COVID, severe infection, obesity, smoking and other conditions.

Please provide questionnaire as a part of the supplements.

Also clarify that the 2 groups are 1. Critical care pts requiring mechanical ventilation and if the other group was critical patients in ICU not requiring MV or were they pts admitted to hospital but not in ICU.

The most compelling question that cannot be answered by this study is the presence of symptoms of PICS and long COVID, the authors need to further highlight that this needs further research and is a limitation of this study. A better clarification as indicated by reviewer 3 of the timing of evaluation would be useful. Change in the analysis of available data to include time to event analysis and symptom free days would make the data more clinically useful.

The detailed review comments and questions raised and included for your reference and for your response.

We look forward to receiving your revised manuscript.

Kind regards,

Shweta Rahul Yemul Golhar, MD

Academic Editor

PLOS ONE

Journal Requirements:

5. One of the noted authors is a group or consortium “Occupational Health and Preventive Medicine Consortium”. In addition to naming the author group, please list the individual authors and affiliations within this group in the acknowledgments section of your manuscript. Please also indicate clearly a lead author for this group along with a contact email address.

Reviewers' comments:

Reviewer's Responses to Questions

**Comments to the Author**

1. Is the manuscript technically sound, and do the data support the conclusions?

Reviewer #1: Partly

Reviewer #2: Yes

Reviewer #3: Partly

2. Has the statistical analysis been performed appropriately and rigorously? 

Reviewer #1: No

Reviewer #2: Yes

Reviewer #3: No

3. Have the authors made all data underlying the findings in their manuscript fully available?

Reviewer #1: No

Reviewer #2: Yes

Reviewer #3: Yes

4. Is the manuscript presented in an intelligible fashion and written in standard English?

Reviewer #1: Yes

Reviewer #2: Yes

Reviewer #3: No

5. Review Comments to the Author

Reviewer #1: The authors followed-up 280 patients that recovered from COVID19 during the period of December 23rd 2020 to April 24th 2021 and reported long-term symptoms associated with COVID19. While this study is important and I would encourage studies like this to be published to get more information on the long-term impact of COVID19 in human health; I would suggest a major revision for this study.

Firstly, I identified a major ethical issue. Study-participants did not sign a consent form even when they went to clinic for evaluation. The authors state that a verbal consent was obtained, but this cannot be verified. Also, data were collected before the authors got an approval from the ethics committee. Good Clinical Practice (GCP) dictates that the study protocol get approval by the institutional review board before any study data are collected or accessed. The researchers need to make sure that their study meets the international ethical standards.

Line 66-68: The percentages are misleading. They refer to certain areas and not the global levels. Needs to be corrected either have percentages that depict the global impact or focus on specific areas of the world.

Line 74: the authors refer to Long-Term COVID-19, please elaborate. Do you mean that these people had COVID-19 infection for long period of time or that the side effects after COVID-19 infection lasted for long time?

Line 93: Please elaborate on the criteria for severe infection. Since these may differ in different parts of the world this need to be defined.

Line 95: For the patients that had the PCR test do you have data on the corona virus strain? If yes it would be interesting to see if these long-term symptoms are associated with certain corona virus strain.

Line 110: Please provide the questionnaire that was used to these patients. Did this questionnaire got approval from the ethics committee before given to the study-participants?

Line 113: Please elaborate on the clinical examination. What did it include? What were the complementary examinations?

Line 116: What was the condition of these patients at baseline before COVID19? Are the symptoms described de novo for these patients?

Line 123: Musculoskeletal symptoms is this due to prolonged hospitalization? Not directly related to COVID19 infection.

Line 127: There is an increase of anxiety and depression cases in global level due to socioeconomical changes that happened during the pandemic. Not directly related to the infection.

Line 166: Oxygen saturation was ranging from 20 to 97%? It is not clear. Did you measure oxygen saturation to only 73.45% of the patients? I am not sure why oxygen saturation hasn’t been reported for all the patients.

Line 171: Only 12.5% of the patients were previously healthy. It would be interesting to see the comparison of the long-term symptoms of COVID19 infection to healthy patients versus the patients with preexisting conditions.

Line 174: I am not sure if this is indeed gender-related. This could be attributed to a high % of men with pre-existing conditions.

Line 193: Gender oriented analysis is needed. Did you see differences in these symptoms between men and women?

Intubated and non-intubated is not the only variable here. I am assuming that the patients that had to be intubated must have had pre-existing conditions, so the differences reported between the two groups could be expected.

What were the treatments these participants received during their hospitalization? Could some of the reported symptoms be attributed to the high-doses of steroid treatments? You could group your patients based on the treatment received and correlate the long-term symptoms after hospitalization.

The authors collected valuable data that could give us an insight on the long-term health impact of COVID19 infection, however its is important to take under consideration the multiple factors that could have contributed to the symptoms described.

Reviewer #2: Summary: Irisson-Mora et al. have conducted an interesting study comparing the long-term sequelae of COVID-19 between hospitalized patients who required mechanical ventilation and hospitalized patients that did not. The authors administered a survey to patients with confirmed COVID at 6 months after discharge that captured 8 domains of symptoms and compared the prevalence of symptoms between groups. They found that mechanically ventilated patients, unsurprisingly, had a very high prevalence of persistent symptoms at 6 months (98%), and had higher odds of developing neuropsychiatric and musculoskeletal symptoms that patients that were not mechanically ventilated. The strength of the study is that seems to be well designed with appropriate statistical analysis. The major limitation of the study is the lack of novelty of the results, which may be rectified by some discussion about how the findings in this cohort compare to what is expected in patients who suffer from post intensive care unit syndrome not due to COVID-19.

Major revisions:

1) While there is some novelty in these findings as the time frame is somewhat longer than other studies of the long-haul COVID and more focus on neuro-psychiatric symptoms, the main conclusions of the study are unsurprising, and many of them have been reported previously (PMID: 34308300, PMID: 34308300). A strength of this is that it increases confidence in the findings, but a major limitation is that it does not seem to add new knowledge to aid in prognosis and understanding the pathophysiology of this phenomenon. Including some discussion comparing how the findings of this cohort differ from what others have found may rectify this somewhat, but if there are no major differences in the findings of this cohort and previous studies, it is difficult to see how this contributes anything novel.

2) Additionally, it’s not clear how many of these persistent symptoms are related to the general phenomenon of post-intensive care unit syndrome (PICS), and how many of these symptoms are related specifically to SARS-CoV-2 infection. The authors acknowledge that many of their reported findings are consistent with the general phenomenon of PICS in lines 258-259, but do not expound on this further. Exploring the differences here would be interesting and increase the novelty of the findings, as differences could give some insight into how SARS-CoV-2 infection specifically impacts and modifies PICS.

3) There has been a lot of work into the development of appropriate survey instruments and outcome measures for the study of PICS (PMID: 34025756; PMID: 30600222; PMID: 30600222). While the domains and measures listed by the authors do seem reasonable, it would be helpful to know more detail about how this survey was developed and whether validated metrics and questionnaires were used, and if not, what validation process and testing went into the questionnaire design. Additionally, including the survey instrument with the questions listed in the supplement would be helpful.

Minor revisions:

1) A description of the criteria for pneumopathy (Lines 176 and 261, and Table 1) would be helpful. Are there formal criteria with pulmonary function testing or is this anyone with a history of lung disease?

2) In Figure 3 legend, specifying the methods used to determine significant differences would be helpful, are the p-values listed for single variable or multivariable analysis?

3) The language in the manuscript is clear but line 146 (“we dumped it into a database”) might benefit from different language that better describes the hard work and care that the authors put into the analysis.

Reviewer #3: This paper asks an important question related to COVID-19: Of the critically ill patients, how many have residual deficits between 2 weeks and 6 months after discharge, and do patients who had received invasive mechanical ventilation (IMV) have more significant residual deficits compared to those who did not receive mechanical ventilation? This is a prospective, single center observational study with what is likely an adequate sample size (although power calculations were not included). However, there are some concerns, especially related to analytical approach, that I would recommend addressing-

Major comments

- It is not clear from the manuscript the timing of reported symptoms (e.g. in page 16 you state "at of evaluation")- are they all present at 6 months? Were they reported at 2 weeks but dissipated by 6 months? Something in between? Assuming some patients reported symptoms at 3 weeks and other at 5 months and everything in between, this lack of temporal resolution leads to less specific findings, which end up being less clinically meaningful. As a physician, I would not know how to use these results to advise patients- will the symptoms that they have reported 2 weeks after discharge persist for much longer? If they don't have symptoms now, will they develop them later? I couldn't tell. Is there a difference in time course depending on what symptom was reported (eg do PTSD last longer than the paresthesias?). I would try to be more specific about time course.

- In addition, given the nature of the measured outcome, I would have used time to event analysis (eg time to symptom resolution) to compare those who received IMV vs. not. I would also include "symptom free days" for chosen symptoms in the analysis for an evaluation with less time and death related confounding. If this is done, please add a time to event graph to the figures as this would be more informative than binary bar graphs.

- If possible, please state whether there is a difference in baseline characteristics between patients excluded from the study vs. those who were included.

- In page 9, when there is a statement about whether IMV was affected by smoking and other co-morbities, please realize that there are several more nuanced analyses in the literature that report that smoking, age, obesity and other comorbidities indeed are associated with higher odds of intubation. These analyses included multivariable analysis while the statement made in this manuscript were evaluated by chi-square only. I would temper the statements re: conclusions from table 1 given the limited analysis provided here.

- There are some claims in the discussion that lack sufficient evidence from the results that I would therefore recommend rephrasing or removing. For example, in page 20, the statements made in lines 313-318 lack any evidence from the result section and lack any citations. I would remove this paragraph or at least significantly shorten it. Other paragraphs utilize several lines reviewing the literature without direct connection to the contents of the manuscript and should also be shortened or removed.

Minor comments

- There are several grammatical and lexicon mistakes that need to be corrected- a repeat issue is the use of "sequels" instead of the correct term, "sequelae". In page, 7, I would avoid the use of the colloquialism "dumped" into a scientific publication. Likewise, in page 20, the phrase "the hole COVID hospitalization picture" (lines 314-315) has a typo and is not specific enough for a scientific journal.

- Please add error bars to all bar graphs.

- Regarding symptoms reported, it would be useful to have a table (can be in supplemental materials) defining the symptoms reported- eg, pneumopathy, fatigue, sweat, diarrhea, constipation, cognitive impairments. Were these all subjectively reported or were there an objective measure for some of them? For neuropathic pain, was there a correlation with placement of A line or other procedures? was it generalized neuropathic pain?

-Likewise, please define risk factors such as smoking history (minimum packs a day? ever or never smoker?), biomass exposure, obesity, and past medical history (were they just extracted from the chart? reported by patients?).

- In line 326-327, usually "tidal volume minimization", which I assume to mean protective mechanical ventilation, will actually decrease oxygenation with the benefit of improved mortality. Decreasing tidal volume will prevent alveolar damage, protect barrier function and mitigate the local release of pro-inflammatory cytokines, but it usually does not improve oxygenation. Please rephrase.

- There is an important point in the discussion re: better outcomes in patients who receive physical therapy (PT). If available, it would be very insightful to report how many patients received PT in the ICU. A comparison in outcomes between those receiving PT v not would be very interesting. A statement comparing the percentage of patients with COVID who received PT vs. historical numbers pre pandemic would be enough to highlight the issues re:access to standard ICU care in this population.

- In the conclusion, or maybe in the discussion, one could also mention the likely importance of lighter sedation goals to outcomes, and cite the appropriate sources.

6. PLOS authors have the option to publish the peer review history of their article (what does this mean?). If published, this will include your full peer review and any attached files.

Reviewer #1: No

Reviewer #2: No

Reviewer #3: **Yes: **Ana Carolina Costa Monteiro

---

## [Author Response · Author response to Decision Letter 0]

14 Jun 2022

Reviewer #1

1.-Firstly, I identified a major ethical issue. Study-participants did not sign a consent form even when they went to clinic for evaluation. The authors state that a verbal consent was obtained, but this cannot be verified. Also, data were collected before the authors got an approval from the ethics committee. Good Clinical Practice (GCP) dictates that the study protocol get approval by the institutional review board before any study data are collected or accessed. The researchers need to make sure that their study meets the international ethical standards.

A: We believe this is a misunderstanding.

All patients discharged from the hospital with COVID-19 diagnosis were followed up by a telephone call, that’s why we described a cohort study. Since patients come from all around the country and due mostly to economic issues, it is impossible for the majority of them to return to this hospital, so the post-COVID clinic calls patients in order to apply a questionary (developed by physicians in this area, not a validated test) to all of discharged ones. During this call, verbal consent is obtained, in order to apply this questionnaire, and if they had any persistent symptoms or associated pathology that requires follow up, they are asked to come back to the hospital, where physical exam and if required, other tests are performed. All data, are collected in the physical chart and electronic data system of our hospital.

Our study consisted in a chart and electronic review (descriptive, transversal and observational), with approval of the Institution Ethics committee (C22-21) before any data was collected. No consent form was required for this kind of study.

Changes have been made in the design and methods section in order to clarify this point. 

2.-Line 66-68: The percentages are misleading. They refer to certain areas and not the global levels. Needs to be corrected either have percentages that depict the global impact or focus on specific areas of the world.

A: The risk of serious illness reported in countries such as China and the United States varies from 12.6 to 23.5%, and the rate of invasive mechanical ventilation (IMV) among these patients ranges 68 from 29.1 to 89.9%.

Changes have been made to the introduction section to clarify this point.

3.- Line 74: the authors refer to Long-Term COVID-19, please elaborate. Do you mean that these people had COVID-19 infection for long period of time or that the side effects after COVID-19 infection lasted for long time?

A: The term prolonged post-COVID-19 describes persistent symptoms following hospitalization from COVID-19 illness. All of our study patients already had a negative polymerase chain reaction (PCR) test for SARS COV2.

Changes have been made to the introduction section to clarify this point.

4.-Line 93: Please elaborate on the criteria for severe infection. Since these may differ in different parts of the world this need to be defined.

A: Criteria for severe disease were those defined by the WHO (SpO 2 <94%, ratio of arterial partial pressure of oxygen to fractional inspired oxygen (PaO 2 /FiO 2 ) <300 mmHg, respiratory rate >30 breaths/min or pulmonary infiltrates >50%).

Changes have been made in the design and methods section in order to clarify this point. 

5.-Line 95: For the patients that had the PCR test do you have data on the corona virus strain? If yes it would be interesting to see if these long-term symptoms are associated with certain corona virus strain.

Please provide the questionnaire that was used to these patients. Did this questionnaire got approval from the ethics committee before given to the study-participants?

A: The determination of the COVID-19 variant which patients were positive to was not performed. Changes have been made in the design and methods section in order to clarify this point. 

As mentioned before, the questionnaire used in these patients was the one used routinely within the Institute, which was also included in the protocol approved by the ethics committee.

The questionnaire was included in complements.

6.-Line 113: Please elaborate on the clinical examination. What did it include? What were the complementary examinations?

A: Patients who attended the medical appointment in person underwent to a complete physical examination with special attention to the symptoms reported by the patient. The complementary examinations included laboratory tests and chest tomography based on the clinical findings evidenced during the examination.

Changes have been made in the design and methods section in order to clarify this point.

7.-Line 116: What was the condition of these patients at baseline before COVID19? Are the symptoms described de novo for these patients?

A: Only new symptoms reported after the patient's discharge were taken into consideration and included in the study.

8.- Line 123: Musculoskeletal symptoms is this due to prolonged hospitalization? Not directly related to COVID19 infection.

A: When performing the univariate and multivariable analysis between musculoskeletal symptoms and prolonged hospitalization time no statistically significant association was demonstrated (P=0.796), so they were directly attributed to COVID-19 infection.

9.- Line 127: There is an increase of anxiety and depression cases in global level due to socioeconomical changes that happened during the pandemic. Not directly related to the infection.

A: It is indisputable that these symptoms can be a hard to tell apart if whether they can be caused directly by COVID-19 or by the pandemic socioeconomical factors directly. However, it is important to emphasize that in our study they were attributed to post-COVID-19 symptoms, as they were referred by patients as post-hospitalization symptoms, being consistent with the current concept of prolonged COVID-19, so it is appropriate to define them within the spectrum of symptoms related to COVID-19.

10.- Line 166: Oxygen saturation was ranging from 20 to 97%? It is not clear. Did you measure oxygen saturation to only 73.45% of the patients? I am not sure why oxygen saturation hasn’t been reported for all the patients.

A: The percentage of 73.45% refers to the mean oxygen saturation measured by pulse oximetry and not to the number of patients. Changes have been made in the results section to clarify this point.

11.- Line 171: Only 12.5% of the patients were previously healthy. It would be interesting to see the comparison of the long-term symptoms of COVID19 infection to healthy patients versus the patients with preexisting conditions.

A: When performing the univariable and multivariable analysis among healthy patients against patients with previously diagnosed comorbidities, it was concluded that patients with comorbidities reported more respiratory (P=0 .037) and musculoskeletal (P=0.034) symptoms. Changes have been made in the results section to clarify this point.

12.-Line 174: I am not sure if this is indeed gender-related. This could be attributed to a high % of men with pre-existing conditions.

A: Although men had a greater tendency to be intubated (P=0.016), longer hospitalization time (P=0.028) and more comorbidities (P=0.031), no statistically significant differences were found in relation to any pre-existing comorbidity.

Changes have been made in the results section to clarify this point.

13.-Line 193: Gender oriented analysis is needed. Did you see differences in these symptoms between men and women?

Intubated and non-intubated is not the only variable here. I am assuming that the patients that had to be intubated must have had pre-existing conditions, so the differences reported between the two groups could be expected.

What were the treatments these participants received during their hospitalization? Could some of the reported symptoms be attributed to the high-doses of steroid treatments? You could group your patients based on the treatment received and correlate the long-term symptoms after hospitalization.

A: When performing the univariate analysis by gender, it was found that women had a greater exposure to biomass (24.2 vs 9.9%, P=0.002), higher prevalence of respiratory symptoms (71.9 vs 59.2, P=0.032), and dermatological symptoms (25.8 vs 9.2%, P=0 .000) compared to men.

Changes have been made in the results section to clarify this point.

We do not have data regarding the treatment received during patients’ hospitalization and its relation with long-term symptoms after COVID-19 infection, therefore they were not included in this study, which represents a limitation of our study and their inclusion would represent the objective from another job.

Reviewer #2

Major revisions:

1.-While there is some novelty in these findings as the time frame is somewhat longer than other studies of the long-haul COVID and more focus on neuro-psychiatric symptoms, the main conclusions of the study are unsurprising, and many of them have been reported previously (PMID: 34308300, PMID: 34308300). A strength of this is that it increases confidence in the findings, but a major limitation is that it does not seem to add new knowledge to aid in prognosis and understanding the pathophysiology of this phenomenon. Including some discussion comparing how the findings of this cohort differ from what others have found may rectify this somewhat, but if there are no major differences in the findings of this cohort and previous studies, it is difficult to see how this contributes anything novel.

A: Thank you for your comment, as you have mentioned, there are other longer studies evaluating the prevalence of COVID-19. However, most of these include patients from community settings, with mild or moderate COVID. The novelty of our study is that, as a reference center, we have a significant number of critically ill patients, a population in which long-term sequelae have not been sufficiently explored.

2.- It’s not clear how many of these persistent symptoms are related to the general phenomenon of post-intensive care unit syndrome (PICS), and how many of these symptoms are related specifically to SARS-CoV-2 infection. The authors acknowledge that many of their reported findings are consistent with the general phenomenon of PICS in lines 258-259, but do not expound on this further. Exploring the differences here would be interesting and increase the novelty of the findings, as differences could give some insight into how SARS-CoV-2 infection specifically impacts and modifies PICS.

A: COVID-19 critical patients may be predisposed to have a greater prevalence of PICS because of longer periods of mechanical ventilation, sedation and the use of steroids and sedative meditation for their treatment. Although PICS is known to be a common syndrome in patients that needed critical care, and there are a significant number of patients that required this kind of care due to COVID, the prevalence of PICS and its definition in patients with COVID is still not yet determined. Such is the case that in 2021 a call has been made to the World Health Organization (WHO) to develop International Classification of Disease Diagnostic Codes for PICS in the “Age of COVID-19” that incentive to care coordination in outpatient settings and to an effective post-acute treatment. 

In the discussion section we have expanded the explanation underlying the mechanisms that may predispose critically ill COVID-19 patients to the persistence of the symptoms we found.

3.- There has been a lot of work into the development of appropriate survey instruments and outcome measures for the study of PICS (PMID: 34025756; PMID: 30600222; PMID: 30600222). While the domains and measures listed by the authors do seem reasonable, it would be helpful to know more detail about how this survey was developed and whether validated metrics and questionnaires were used, and if not, what validation process and testing went into the questionnaire design. Additionally, including the survey instrument with the questions listed in the supplement would be helpful.

A: The questionnaire used in these patients is the one that has been taken out routinely within the Institute since the start of the pandemic, and is included in the protocol approved by the ethics committee for inclusion in this study, available in supplements.

The domains captured in this tool were categorized into the following 8 symptom classes, depending on the organ system affected: [1] Respiratory symptoms, including dyspnea, chest pain, cough, sore throat, rhinorrhea, wheezing, vocal cord abnormalities ( hoarseness, chordal paralysis) and tracheal stenosis. [2] Cardiovascular symptoms such as clubbing, rhythm disturbances (palpitations, tachycardia, and bradycardia), heart murmur, pericarditis, elevated blood pressure, extremity edema, and deep inferior vein thrombosis. [3] Systemic symptoms, such as fever, dizziness, fatigue, asthenia, adynamia, sweating, pruritus, sensitivity to cold, and tremors. [4]. Musculoskeletal symptoms, such as paresthesias, mobility limitations, arthralgias, tendon injuries, myalgias, paresthesia, neuralgia, muscle weakness, low back pain and contractures. [5] Dermatologic symptoms, including hair loss, skin rashes, and pressure ulcers. [6] Neurological and neuropsychiatric symptoms, such as headache, sleep and wake disorders, cognitive disturbances such as poor concentration and memory loss, as well as anxiety disorders and/or depression. [7] Gastrointestinal symptoms, including nausea, diarrhea, constipation, abdominal pain, and lack of appetite. [8] Audiological symptoms, including tinnitus, ear pain, hearing loss, and vertigo. The impact of each domain on the participants' daily function was reported as present or absent.

Minor revisions:

1.- A description of the criteria for pneumopathy (Lines 176 and 261, and Table 1) would be helpful. Are there formal criteria with pulmonary function testing or is this anyone with a history of lung disease?

A: Included comorbidities were those previously diseases documented in the chart and self-reported by patients; pneumopathy was defined as a history of asthma, chronic obstructive pulmonary disease (COPD) and pulmonary fibrosis.

2.- In Figure 3 legend, specifying the methods used to determine significant differences would be helpful, are the p-values listed for single variable or multivariable analysis?

A: A univariate association analysis was performed using the Chi-square test or Fisher's exact test considering significant P values those less than 0.05. comparing intubated and non-intubated patients. 

Changes have been made in the Figure legend to clarify this point.

3.- The language in the manuscript is clear but line 146 (“we dumped it into a database”) might benefit from different language that better describes the hard work and care that the authors put into the analysis.

A: The word “we dumped” was changed to “were collected on an Excel page”.

Changes have been made in the design and methods section in order to clarify this point. 

Reviewer #3

Major comments:

1.- It is not clear from the manuscript the timing of reported symptoms (e.g. in page 16 you state "at of evaluation")- are they all present at 6 months? Were they reported at 2 weeks but dissipated by 6 months? Something in between? Assuming some patients reported symptoms at 3 weeks and other at 5 months and everything in between, this lack of temporal resolution leads to less specific findings, which end up being less clinically meaningful. As a physician, I would not know how to use these results to advise patients- will the symptoms that they have reported 2 weeks after discharge persist for much longer? If they don't have symptoms now, will they develop them later? I couldn't tell. Is there a difference in time course depending on what symptom was reported (eg do PTSD last longer than the paresthesias?). I would try to be more specific about time course.

A: We carried out a transversal study including persistent symptoms in the period of time that they attended the first medical evaluation after hospital discharge and that were present up to six months after discharge, these symptoms were included within the definition Late sequelae of SARS CoV-2 infection by lasting more than 4 weeks after initial infection or diagnosis, being consistent with the concept of prolonged COVID.

2.- In addition, given the nature of the measured outcome, I would have used time to event analysis (eg time to symptom resolution) to compare those who received IMV vs. not. I would also include "symptom free days" for chosen symptoms in the analysis for an evaluation with less time and death related confounding. If this is done, please add a time to event graph to the figures as this would be more informative than binary bar graphs.

A: Being our study a transversal study, we are not measuring the time from the event to the resolution of the symptoms. It would be interesting to carry out another study looking for the differences between the patients. However, its inclusion would represent the object of another study.

3.- If possible, please state whether there is a difference in baseline characteristics between patients excluded from the study vs. those who were included.

A: For logistical reasons, baseline characteristics among patients excluded from the study were not included.

4.- In page 9, when there is a statement about whether IMV was affected by smoking and other co-morbities, please realize that there are several more nuanced analyses in the literature that report that smoking, age, obesity and other comorbidities indeed are associated with higher odds of intubation. These analyses included multivariable analysis while the statement made in this manuscript were evaluated by chi-square only. I would temper the statements re: conclusions from table 1 given the limited analysis provided here.

A: When performing the univariable and multivariable analysis regarding comorbidities (P=.467) and risk factors such as smoking (P=.454), biomass exposure (P=.130) and alcoholism (P=.155) comparing intubated patients with non-intubated patients, no statistical significance was found.

5.- There are some claims in the discussion that lack sufficient evidence from the results that I would therefore recommend rephrasing or removing. For example, in page 20, the statements made in lines 313-318 lack any evidence from the result section and lack any citations. I would remove this paragraph or at least significantly shorten it. Other paragraphs utilize several lines reviewing the literature without direct connection to the contents of the manuscript and should also be shortened or removed.

A: Thank you for your comments. Added quotes from statements in discussion and reworded paragraph have been made.

Minor comments:

1.- There are several grammatical and lexicon mistakes that need to be corrected- a repeat issue is the use of "sequels" instead of the correct term, "sequelae". In page, 7, I would avoid the use of the colloquialism "dumped" into a scientific publication. Likewise, in page 20, the phrase "the hole COVID hospitalization picture" (lines 314-315) has a typo and is not specific enough for a scientific journal.

A: The corresponding grammatical corrections were made, from “sequels” to “sequelae”.

The word "we dumped" was changed to “were collected on an Excel page".

2.- Please add error bars to all bar graphs.

A: We added error bars to all charts.

3.- Regarding symptoms reported, it would be useful to have a table (can be in supplemental materials) defining the symptoms reported- eg, pneumopathy, fatigue, sweat, diarrhea, constipation, cognitive impairments. Were these all subjectively reported or were there an objective measure for some of them? For neuropathic pain, was there a correlation with placement of A line or other procedures? was it generalized neuropathic pain?

A: All signs and symptoms included were reported subjectively by the patients, the only objective measure being the physical examination performed during the medical evaluation.

4. Likewise, please define risk factors such as smoking history (minimum packs a day? ever or never smoker?), biomass exposure, obesity, and past medical history (were they just extracted from the chart? reported by patients?).

A: Risk factors such as smoking, exposure to biomass and alcoholism were reported as recorded in the patient's file and reported as present or absent.

5. In line 326-327, usually "tidal volume minimization", which I assume to mean protective mechanical ventilation, will actually decrease oxygenation with the benefit of improved mortality. Decreasing tidal volume will prevent alveolar damage, protect barrier function and mitigate the local release of pro-inflammatory cytokines, but it usually does not improve oxygenation. Please rephrase.

A: Some suggested acute-phase interventions for neurological protection include tidal volume minimization in ventilation setting strategies to improve cerebral blood flow and lower cerebral vascular resistance along with less inflammation.

The statement about minimizing tidal volume as a maneuver to improve mortality has been rephrased in the discussion.

6.- There is an important point in the discussion re: better outcomes in patients who receive physical therapy (PT). If available, it would be very insightful to report how many patients received PT in the ICU. A comparison in outcomes between those receiving PT v not would be very interesting. A statement comparing the percentage of patients with COVID who received PT vs. historical numbers pre pandemic would be enough to highlight the issues re:access to standard ICU care in this population.

A: Although it would be very enriching to include variables on in-hospital management, including physiotherapy during hospitalization, this study did not include this information since its objective was to describe persistent symptoms after hospital discharge and not the effect of interventions during hospitalization.

7.- In the conclusion, or maybe in the discussion, one could also mention the likely importance of lighter sedation goals to outcomes, and cite the appropriate sources.

A: The use of some sedative agents such as dexmedetomidine instead of benzodiazepines, less use of continuous deep sedation, along with and a conservative fluid management have also been proposed as useful interventions to maximize neurological function.

In the discussion section we have expanded the explanation to clarify this point.

---

## [Decision Letter · Decision Letter 1]

1 Jul 2022

PONE-D-21-28914R1Comparison between the persistence of post COVID-19 symptoms on critical patients requiring invasive mechanical ventilation and non-critical patients.PLOS ONE

Dear Dr. Miguel Angel Salazar-Lezama,<table border="0" cellpadding="0" cellspacing="0" class="datatable3" style="border-collapse: collapse; width: 677px; line-height: 14px; caret-color: rgb(0, 0, 51); color: rgb(0, 0, 51); font-family: verdana, geneva, arial, helvetica, sans-serif; font-size: 11.199999809265137px;"> 

</table>

Thank you for submitting your manuscript to PLOS ONE. After careful consideration, we feel that it has merit but does not fully meet PLOS ONE’s publication criteria as it currently stands. Therefore, we invite you to submit a revised version of the manuscript that addresses the points raised during the review process.

The authors have addressed most of the concerns raised by myself and the reviewers. The aspect of limitations of the study as raised by the reviewers is vital to bring forth so the reader understands the shortcomings of the assessment, hence I would advise the authors to add these limitations as requested by reviewer 3.

We look forward to receiving your revised manuscript.

Kind regards,

Shweta Rahul Yemul Golhar, MD

Academic Editor

PLOS ONE

Journal Requirements:

Reviewers' comments:

Reviewer's Responses to Questions

**Comments to the Author**

1. If the authors have adequately addressed your comments raised in a previous round of review and you feel that this manuscript is now acceptable for publication, you may indicate that here to bypass the “Comments to the Author” section, enter your conflict of interest statement in the “Confidential to Editor” section, and submit your "Accept" recommendation.

Reviewer #2: All comments have been addressed

Reviewer #3: (No Response)

2. Is the manuscript technically sound, and do the data support the conclusions?

Reviewer #2: Yes

Reviewer #3: Yes

3. Has the statistical analysis been performed appropriately and rigorously? 

Reviewer #2: Yes

Reviewer #3: Yes

4. Have the authors made all data underlying the findings in their manuscript fully available?

Reviewer #2: Yes

Reviewer #3: Yes

5. Is the manuscript presented in an intelligible fashion and written in standard English?

Reviewer #2: Yes

Reviewer #3: Yes

6. Review Comments to the Author

Reviewer #2: The authors seem to have addressed most of my concerns, and I appreciated them adding the survey instrument to the supplement. I believe that the analysis and study design are technically sound.

Reviewer #3: Thank you for all your thorough responses to the comments. I would encourage the authors to add a paragraph in the discussion covering the limitations that could not be addressed by this study- namely, inability to discern a specific time line of symptoms as there was only one time point recorded, the fact that all symptoms were patient reported and lacked objective criteria for determination, and that this was an observational study with inherent bias regarding how symptoms were reported.

7. PLOS authors have the option to publish the peer review history of their article (what does this mean?). If published, this will include your full peer review and any attached files.

Reviewer #2: No

Reviewer #3: No

---

## [Author Response · Author response to Decision Letter 1]

21 Jul 2022

We welcome comments raised by reviewers. The limitations of this study have already been added in the discussion section and the list of references was checked.

---

## [Editor Report · Decision Letter 2]

2 Aug 2022

Comparison between the persistence of post COVID-19 symptoms on critical patients requiring invasive mechanical ventilation and non-critical patients.

PONE-D-21-28914R2

Dear Dr. Miguel Ángel Salazar-Lezama,

We’re pleased to inform you that your manuscript has been judged scientifically suitable for publication and will be formally accepted for publication once it meets all outstanding technical requirements.

Kind regards,

Shweta Rahul Yemul Golhar, MD

Academic Editor

PLOS ONE

Additional Editor Comments (optional):

I congratulate the authors on researching the complicated and challenging topic of long COVID. The research on this topic adds to our ever increasing knowledge and provides information to be able to manage the immense burden projected for years to come for healthcare. As we define the condition better and try to differentiate other chronic conditions like PICS, the similarities and differences should get clearer, helping us provide more specific care and better outcomes.
---

## [Editor Report · Acceptance letter]

5 Aug 2022

PONE-D-21-28914R2 

Comparison between the persistence of post COVID-19 symptoms on critical patients requiring invasive mechanical ventilation and non-critical patients. 

Dear Dr. Salazar-Lezama:

I'm pleased to inform you that your manuscript has been deemed suitable for publication in PLOS ONE. Congratulations! Your manuscript is now with our production department. 

Kind regards, 

on behalf of

Dr. Shweta Rahul Yemul Golhar 

Academic Editor

PLOS ONE